# Experimental Infection of Ticks: An Essential Tool for the Analysis of *Babesia* Species Biology and Transmission

**DOI:** 10.3390/pathogens10111403

**Published:** 2021-10-29

**Authors:** Sarah I. Bonnet, Clémence Nadal

**Affiliations:** 1Animal Health Department, INRAE, 37380 Nouzilly, France; 2Functional Genetics of Infectious Diseases Unit, Institut Pasteur, CNRS UMR 2000, Université de Paris, 75015 Paris, France; 3Epidemiology Unit, Laboratory for Animal Health, University Paris Est, 94700 Maisons-Alfort, France; clemence.nadal@anses.fr; 4Anses, INRAE, Ecole Nationale Vétérinaire d’Alfort, UMR BIPAR, Laboratoire de Santé Animale, 94700 Maisons-Alfort, France

**Keywords:** ticks, *Babesia* sp., biological cycle, experimental transmission, experimental models

## Abstract

Babesiosis is one of the most important tick-borne diseases in veterinary health, impacting mainly cattle, equidae, and canidae, and limiting the development of livestock industries worldwide. In humans, babesiosis is considered to be an emerging disease mostly due to *Babesia divergens* in Europe and *Babesia microti* in America. Despite this importance, our knowledge of *Babesia* sp. transmission by ticks is incomplete. The complexity of vectorial systems involving the vector, vertebrate host, and pathogen, as well as the complex feeding biology of ticks, may be part of the reason for the existing gaps in our knowledge. Indeed, this complexity renders the implementation of experimental systems that are as close as possible to natural conditions and allowing the study of tick-host-parasite interactions, quite difficult. However, it is unlikely that the development of more effective and sustainable control measures against babesiosis will emerge unless significant progress can be made in understanding this tripartite relationship. The various methods used to date to achieve tick transmission of *Babesia* spp. of medical and veterinary importance under experimental conditions are reviewed and discussed here.

## 1. Introduction

Babesiosis remains prevalent worldwide and represents an important threat for both humans and animals [1,2]. The disease, impacting mainly cattle, sheep, goat, equidae, canidaecanidae, and accidentally humans, is caused by apicomplexan parasites belonging to the *Babesia* genus that exclusively infect erythrocytes of their vertebrate hosts [3]. To date, more than 100 *Babesia* species have been identified [1]. *Babesia* spp. are transmitted by hard ticks—occasionally by blood transfusion—and require both a competent vertebrate and invertebrate host to maintain the transmission cycle [3]. 

Human babesiosis is caused by *Babesia microti,* a *Babesia crassa*-like pathogen, *Babesia divergens, Babesia duncani,* and *Babesia venatorum*, as well as other parasites closely genetically related to these pathogens, such as *B. divergens*-like, *B. duncani*-like, and *B. microti*-like [4]. Infections in otherwise healthy individuals is usually mild to moderate and most cases of severe disease occur in immunocompromised individuals. *B. microti* is endemic in the northeastern and upper midwestern regions of the United States, while *B. duncani* is present on the west coast of the country [4]. In Europe, most of the human cases are due to *B. divergens,* whereas in Asia they are due to *B. venatorum, B. crassa*-like, and *B. microti* [4]. In cattle, *Babesia* spp. have a significant worldwide economic, social, and epidemiological impact and include, among the most important species, *B. bovis, B. bigemina, Babesia major,* and *B. divergens* [5]. *B. bovis* and *B. bigemina* are present in many countries in Africa, Asia, Australia, Central and South America, and Southern Europe between 40° N and 32° S. *B. major* is present in Europe, Northwest Africa and Asia, and *B. divergens* is present in northern Europe [5]. Ovine babesiosis due to *Babesia ovis* and *Babesia motasi* is considered as the most critical blood-borne parasitic disease of small ruminants in tropical and non-tropical regions (occurring in South-eastern Europe, North Africa, and Asia) [6]. In equids, *Babesia caballi* is (with *Theileria equi* and *Theileria haneyi*) the agent of equine piroplasmosis known to be endemic in several countries of Africa, Asia, the Americas, and mainly in the Mediterranean basin for Europe [7,8,9]. The disease represents a significant animal health issue and causes notable economic losses for the equine industry. Finally, babesiosis is one of the most important globally extended and quickly spreading tick-borne diseases in dogs worldwide. *Babesia canis* is the main cause of canine babesiosis in Europe and is only sporadically found around the world, whereas *Babesia gibsoni,* the most prevalent species, and *Babesia vogeli* have a global distribution. *Babesia rossi*, one of the most pathogenic species, is endemic in southern Africa [10]. 

The current approaches available for babesiosis control have many important limitations, including increased resistance to acaricides by ticks, as well as the numerous drawbacks of these acaricides and of the current vaccines and babesicidal drugs (e.g., efficacy, toxicity, environmental effects) [11]. The development of improved control measures against babesiosis is limited by the numerous and significant gaps in our understanding of the biology of *Babesia* spp., especially regarding molecular interaction between parasites, vectors, and vertebrate hosts, as well as the factors that may influence both the development and the transmission of the parasite [12]. To fill these gaps, it is essential to be able to reproduce the life cycle of *Babesia* species in controlled experimental conditions, including transmission by ticks [13,14]. In addition, the validation of new methods of interruption of the cycle requires that it can be first carried out entirely under such conditions. Finally, it is important to mention that despite the promise of in vitro culture systems [15], maintenance by in vitro culture [16] or needle-passage in the vertebrate host [17] in the absence of tick passage may generate significant changes in the parasite population, potentially creating a bias in research results.

Several laboratory studies that aimed to understand babesiosis pathogenesis have focused their interest on infecting laboratory animals through artificial parasite injection [18,19]. Regarding tick-parasite interaction, the great majority of studies carried out concern mainly epidemiological studies focusing on the detection of parasites in ticks collected into the field [20]. Some in vitro studies have also been performed in order to understand the interactions between parasites and the cells of their vertebrate [21,22,23] or invertebrate hosts [24,25,26,27]. However, and certainly because of the difficulties inherent to the studied model—including *Babesia* spp. culture, tick colony maintenance, and animal models—relatively few studies have been based on the establishment, under experimental conditions, of complete parasite transmission cycles from one vertebrate host to another via the tick bite. Thanks to the experimental models developed for that purpose, these few studies have nevertheless made significant advances in (1) the definition/confirmation of the vector competence of various tick species; (2) the understanding of the modalities of parasite acquisition and transmission by ticks; (3) the discovery of the molecular interactions between the parasite and its invertebrate hosts; (4) the evaluation of some control methods. The aim of this review is to summarize studies that include both tick infection on *Babesia*-infected animals and *Babesia* infection of ticks through artificial systems, and to comment on the major results they achieved. 

## 2. General Description of the *Babesia* Life Cycle

*Babesia* spp. are transmitted by hard tick (ixodid) vectors. The tick vectors and reservoir hosts differ depending on *Babesia* sp. and geographical location considered [1]. Several tick species have been mentioned in the literature as vectors of *Babesia* sp., but, as shown in Table 1, vector competence through experimental transmission has not been validated for all of them. For those mentioned here as "suspected vectors" without realization of the complete transmission cycle under experimental conditions, their involvement is mostly based on epidemiological evidence (e.g., correlation of tick species presence with disease occurrence). Each of the three active stages of hard ticks (larva, nymph, and adult) takes a single blood meal from a vertebrate host in order to mature to the next stage or lay eggs for the female. Most of the tick species listed as confirmed or suspected vectors of some *Babesia* species are three-hosts ticks—meaning that they take their three blood meals on three different hosts. Some of them, however, have a two-host cycle such as some *Hyalomma* spp., *Rhipicephalus evertsi,* and *Rhipicephalus bursa,* whereas *Dermacentor nitens, Rhipicephalus *annulatus,* Rhipicephalus microplus,* and *Rhipicephalus decoloratus* are one-host ticks that use the same individual host animal for all active tick stages.

The *Babesia* life cycle includes both asexual multiplication in the erythrocytes of the vertebrate host and sexual reproduction in the tick vector [3,61]. The general life cycle of the *Babesia* species is summarized in Figure 1 for *Babesia* sensu stricto (s.s.) species. Indeed, it is necessary to specify here that, quite recently, molecular phylogeny studies using the 18S rRNA gene have led to the division of *Babesia* species into two large groups: *Babesia* s.s. and *Babesia* sensu lato (s.l.). Species belonging to the latter group, such as *B. microti*, are not capable of transovarial transmission within the tick but only a transstadial mode of transmission [62,63]. Vertebrate hosts are infected by the injection of sporozoites present in tick saliva during the tick bite. Each sporozoite penetrates the cell membrane of an erythrocyte with the aid of a specialized apical complex. Once inside, the parasite produces two merozoites by a process of merogony. Merozoites are then intermittently released following erythrocyte lysis to infect new erythrocytes. The parasite may then persist asymptomatically within its host for several years or lead to acute disease. When they are ingested by the tick during the blood meal, some parasites present in infected erythrocytes (pre-gametocytes) undergo further development in the passage from host blood to the midgut of the tick vector to evolve into gametocytes. The sexual reproduction between gametocytes takes place in the tick gut and leads to a zygote that penetrates the gut epithelium, where further multiplication occurs, with development to motile and haploid kinetes that escape into the tick hemolymph. The kinetes then infect a variety of tick cell types and tissues, including the ovary in the female tick—for *Babesia* s.s. species—and the tick salivary glands, where successive cycles of asexual multiplication take place. In this last organ, sporozoite development usually only begins when the infected tick attaches to the vertebrate host. The ticks thus transmit the sporozoites to a new host during a new blood meal of the next life-stage for the ticks with several hosts or of the next generation after transovarian transmission for the one-host ticks. 

## 3. Experimental Models of *Babesia* Species—Transmission by Ticks

### 3.1. Tick Infestations on Babesia spp. Infected Animals

The first experiments to transmit *Babesia* spp. naturally were carried out by applying ticks suspected of being vectors on infected animals. Due to the huge economic importance of bovine babesiosis, these studies on vector competence for *Babesia* sp. were first conducted on species that infect cattle. Indeed, in 1893, two American researchers, Smith and Kilborne—the first authors to demonstrate the transmission of a disease organism from an arthropod to a mammalian host—showed the vector competence of *R. *annulatus** for *B. bigemina* by placing ticks infected on animal onto naïve cattle that developed the associated disease [42]. Thereafter, different species of animals were used depending on the species of *Babesia* studied. Over time, the animal models and methodology used were refined to optimize the infection of both animals and ticks, and to comply with health and safety rules, and animal accommodation and tick containment methods during the tick feeding process have been the subject of several tests and evolutions (see examples in Figure 2A–F). In most cases, for cattle, animals were housed in individual, tick-proof pens surrounded by moats with or without detergent or insecticide [43,45]. Concerning the tick containment methods, in 1961, Callow and Hoyte used a hessian rug to protect the larvae until adult repletion to demonstrate the transmission of *B. bigemina* to cattle by *R. microplus* [43]. Ticks were either allowed to spread at will over the animal, or were confined to one site by releasing them under a fabric patch (nylon or organdie), which was glued along its edges to the flank of the bovine. A few years later, for the demonstration of the vector competence of *R. decoloratus* for *B. bigemina* in Kenya, ticks were fed on cattle until adult repletion by sprinkling larvae on the backs of the animals [45]. For the first experimental transmission of *B. divergens* by *Ixodes ricinus* achieved by Joyner et al in 1963, the ticks were contained in ear bags [36], a method also used later for sheep [64]. Regarding rodents, several laboratory studies involving complete transmission cycles of *B. microti* to the vertebrate host through the tick bite were performed [28,29,30,31,32,33]. In most instances, rodents were maintained over trays of water from which detached engorged ticks—applied by brush to animals—were harvested, whereas in some cases, ticks were contained in plastic capsules attached with different adhesives. For horses, the first studies aimed to validate the vector competence of *D. nitens* for *B. caballi* used larvae applied by brush on the animal [51], whereas in subsequent studies, tick feedings were accomplished by placing the larvae under a cloth patch glued to the back of the host [52].

The identification of a pathogen or pathogen DNA alone—which is even less convincing—in an arthropod cannot be sufficient to prove its ability to transmit this pathogen. Indeed, demonstrations of parasite presence in unfed field ticks, in tick salivary glands, eggs, or unfed larvae, while more convincing than detection in ticks collected from animals, also require confirmation only provided by the validation of vector competence in a controlled experimental model. The best illustration of this corresponds to the following studies performed in Iran in order to identify the vector of *B. ovis* to sheep. The kinetes of *B. ovis* were observed in hemolymph and egg smears of *Rhipicephalus sanguineus* and *Hyalomma marginatum* field ticks collected from sheep infected with *B. ovis* [69], so the vector competence of both tick species was further evaluated by placing pairs of adult ticks on sheep inoculated with *B. ovis,* but no transmission by any of the succeeding tick stages could be demonstrated, thus showing that these tick species are not vectors [64]. 

As recently reviewed by Gray and co-workers, the establishment of experimental models of pathogens transmission by ticks using live animals has led to significant advances in the understanding of transmission modalities and tick-parasite interactions [70]. For example, it allowed the demonstration that transovarial transmission does not occur in *B. microti,* leading to no longer classifying this parasite in *Babesia* spp. s.s. [31,32,33], and that, in *I. ricinus,* the infection only survives one molt [31]. Laboratory models have also demonstrated that *B. microti* may promote its transmission in rodents by enhancing the feeding success and survival of its tick vector, *Ixodes trianguliceps* [29]. Likewise, the establishment of the transmission of *B. divergens* by *I. ricinus* in a gerbil experimental model [35] provided proof of sexual development of *Babesia* through DNA measurements on the developmental stages of *B. divergens* in the blood of the vertebrate host and in the gut, hemolymph, and salivary glands of the tick vector [38,39]. In addition, several studies have been performed in order to establish which tick life stages are able to acquire and/or to transmit the parasite. Most of them have concluded that only adult stages were able to acquire *Babesia* sp. s.s. from infected animals, while all succeeding stages (larvae, nymphs, and adults) were able to re-transmit the parasites to susceptible animals [36,37,43,44,48]. However, Schwint and co-workers demonstrated that only the first of three subsequent generations from *D. nitens* females was able to transmit *B. caballi* to naïve horses, showing that the parasite is unable to persist in ticks without continuing alimentary infection of adult females [52], whereas other studies have also shown the acquisition of this parasite by nymphs of *R. evertsi* [53]. 

Models of transmission of *B. microti* to rodents via ticks in the laboratory have also made it possible to carry out studies on the phenomena of co-infections. In fact, a lower transmission efficiency of *B. microti* than *Borrelia burgdorferi* to *Ixodes dammini* from both hamsters [71] and white-footed mice [72] has been demonstrated. In the meantime, it was shown that ticks that fed on mice with these concurrent pathogen infections exhibited twice the incidence of *B. burgdorferi* infection compared with *B. microti* [72]. Twenty-five years later, however, this laboratory model also showed an increase of the frequency of *B. microti*-infected *Ixodes scapularis* (formerly *I. dammini*) nymphs when they fed as larvae on white-footed mice coinfected with *B. burgdorferi,* as well as an increase of *B. microti* parasitemia in co-infected mice [28]. This enhancement of *B. microti* establishment by *B. burgdorferi* has been attributed to an immunological conflict in the adaptive immune response of the vertebrate host against the two tick-borne pathogens [73]. 

The development of experimental models using animals also provided knowledge on the infection acquisition by the vertebrate host following a *Babesia*-infected tick bite. For example, by developing a laboratory model of *B. bovis* infection of calves through the bite of *R. microplus*-infected ticks, Smith and co-workers demonstrated, in 1978, that tick-induced infection was more severe than in calves infected with carrier blood, even when very low numbers of infected larvae were applied [41]. They attributed this difference in virulence to the large number of infective doses injected by each infected tick but Salivary-Assisted Transmission of tick-borne pathogens (see review in [74]) probably also contributed to this observation. 

Although, compared to other pathogens, few molecular studies have involved *Babesia* parasites [70,75], experimental tick-transmission models of *Babesia* spp. using animals have also made it possible to identify molecules potentially involved in this transmission. Such studies are helping us to better understand the interactions involved and to identify potential targets for blocking parasite transmission. Thus, infection of *H. longicornis* on dogs infected with *B. gibsoni* has allowed to implicate a tick protein, longipain, in the transmission of the parasite by its vector [76], and to demonstrate that the vitellogenin receptor on the surface of tick oocytes is essential for its transovarial transmission [77]. Experimental models of *B. bovis* infection of cattle by *R. microplus* has also been used to study the function of the tick protein Bm86 during *B. bovis* infection [78]. In the same way, experimental tick infection models of both *B. bigemina* and *B. bovis*-infected cattle were used to perform functional genomics studies on *R. annulatus* and *R. microplus* genes that are differentially expressed in response to parasite infection [79,80]. 

Both the discovery and validation of methods to control the transmission of tick-borne pathogens require experimental designs that include complete transmission cycles. Regarding the *Babesia* spp. that infect cattle, experimental tick infection models for *B. bigemina*-infected cattle were used to evaluate the efficiency of some vaccine candidates and drugs against parasite transmission by *R. microplus* [81,82]. *Rhipicephalus microplus* experimentally infected with *B. bovis* were also used to demonstrate the inefficiency of the injectable and pour-on forms of both ivermectin and moxidectin to prevent parasite transmission by ticks [82]. In Argentina, Mangold and co-workers developed a laboratory model of *B. bovis* transmission to cattle by *R. microplus* in order to demonstrate the non-transmissibility to ticks of an attenuated vaccine strain of the parasite [83]. Several studies also involved experimental transmission of *B. canis* to dogs through the bite of infected *D. reticulatus* ticks in order to evaluate the usefulness of different acaricides to prevent parasite transmission [55,56,57,84], whereas similar experiments were performed regarding *B. canis* transmission by *R. sanguineus* [85,86], and *B. rossi* transmission by *Haemaphysalis elliptica* [60]. 

### 3.2. Tick Infection through Artificial Feeding Systems 

The use of natural hosts for direct infection of ticks on infectious animals remains the best method to obtain conditions that are closest to the physiological reality of tick-borne pathogen transmission. However, firstly, ethical considerations lead us to limit the use of animals as much as possible. Secondly, in addition to the constraints associated with licensing the experiments and host specificity, obtaining the animals, keeping them in the laboratory, and handling them can be expensive and difficult or even impossible in the case of most species of wildlife. Finally, the difficulty of controlling parasitaemias of infected animals for the whole feeding period of the ticks is an important consideration. All these reasons have led to the development of artificial methods of tick infection in order to complete the lifecycle of tick-borne pathogens under laboratory conditions. Artificial feeding of ticks, mimicking the natural process, has been used for different purposes including tick rearing, the study of tick physiology and the effects of antibodies or drugs on tick physiology, functional genomic studies, and vaccine candidate discovery, as well as the analysis of tick-borne pathogen transmission (see review by Bonnet and Liu [13]). Nevertheless, relatively few studies have involved *Babesia* parasites. One of the factors limiting these studies is undoubtedly the need for successful in vitro cultivation of the *Babesia* species of interest [15].

The use of blood-filled capillary tubes placed over the mouthparts of ticks was first reported in 1938 by Gregson, who used this technique to collect saliva from *D. andersoni* [87]. Since then, this technique has been used to infect ticks with several tick-borne pathogens, mainly bacteria (see review in [13]). Although this infection process has the advantage of using the natural route of infection, i.e., the digestive tract of the tick, and allows control of the quantity of pathogen ingested, it is quite far from natural conditions because the tick absorbs a large amount of pathogen at once and this, regardless of the "true" full blood meal. The other drawback is that it requires the use of animals before or after infection to feed ticks to repletion. However, its use has led to some significant advances in the study of *Babesia* spp. transmission. In 1998, Inokuma and Kemp used the capillary feeding technique in order to infect *R. microplus* with *B. bigemina* [88]. Adult ticks were pre-fed on cattle and then *B. bigemina*-infected red blood cells were offered to the ticks using glass capillaries on a warm plate at 35 °C for 18 h in the dark. The authors subsequently demonstrated that ticks were able to acquire the parasite and to transmit it to their progeny. The capillary feeding technique has also been used by Antunes and co-workers in order to evaluate the impact of purified rabbit polyclonal antibodies on some *R. microplus* proteins associated with both tick physiology and tick infection by *B. bigemina* [89]. While an effect on tick weight and oviposition was observed, no effect was observed on pathogen DNA levels. However, this work has made it possible to set up an alternative system to the use of animals both to test vaccine candidates and to obtain essential data on tick-pathogen interaction, as this could also be done later with another protein, the calreticulin, identified as being involved in *B. bigemina* infection in *R. annulatus* ticks [90].

The membrane feeding technique—consisting of feeding ticks on blood or culture media through a membrane—was first developed by Pierce and Pierce in 1956 in order to feed *R. microplus* using embryonated hen eggs [91]. Since then, several membranes of animal or artificial origin have been used both to feed and to infect ticks with different pathogens (see review in [13]). In this case, as the pathogen is mixed in blood and absorbed throughout the blood meal via the digestive tract, the method mimics the natural conditions of tick infection more closely than other methods. Its main disadvantages, however, are the need to regularly change the blood used, and the need to use antibiotics and antifungals to avoid contamination. It is also necessary to test pathogen viability under these feeding conditions at regular intervals. In the case of artificial membranes such as silicone ones, feeding systems also need olfactory stimuli for attachment and feeding. Regarding their use to infect ticks with *Babesia* spp., and due to red blood cell sedimentation, it is important to note that this requires the placing of blood above the membrane to produce a continuous gravitational pressure, ensuring tick absorption of the intraerythrocytic parasites. In 2007, Bonnet and co-workers developed a skin-feeding technique using the skins of both gerbils (for larvae and nymphs) and rabbits (for nymphs and adults) to infect *I. ricinus* with *B. divergens* without the need for additional stimuli (Figure 2G,H) [35]. To our knowledge, this is the only membrane-feeding technique that has been used to date to infect ticks with *Babesia* spp. All tick instars were allowed to acquire the parasite from infected red blood cells maintained at 37 °C in a glass feeder through the animal skin until repletion. Contrary to what was previously observed [36,37], this system showed that in addition to adult females, *I. ricinus* larvae and nymphs can also acquire *B. divergens* infections, which persists transtadially in the subsequent nymphal and adult stages (as determined by the detection of DNA in their salivary glands). *Babesia divergens* DNA was also detected in eggs and larvae produced by females that had fed on parasitized blood, demonstrating the transovarial transmission of the parasite. Later, the use of this artificial tick infection system also allowed the discovery of molecular markers for *B. divergens* sexual stages [92,93]. Lastly, the same membrane feeding technique allowed the validation of the vector competence of *I. ricinus* for *Babesia* (EU1) *venatorum* [40]. 

Finally, in order to understand tick-*Babesia* interactions and to follow parasite development in the vector, Maeda and co-workers used a “semi-artificial” mouse skin membrane feeding technique to infect *H. longicornis* with *B. ovata* [94]. In this case, female adult ticks were first allowed to feed on the shaved back of mice, and after 4 to 5 days, a section of the mouse skin with the ticks attached was removed immediately after euthanasia, and set up in artificial feeding units. The ticks were then fed on a mix of media and *B. ovata*-infected red blood cells through the piece of mouse skin. This technique was then used to demonstrate the transovarial persistence of *B. ovata* DNA in *H. longicornis* [50]. Thus, the mixing of animal use and membrane feeding makes it possible to control the parasitemia of the meal offered to ticks, but does not prevent the use of live animals.

### 3.3. Tick Infection through Injection 

Although this method is more distant from physiological reality than the ones previously detailed, some studies have also performed tick infections by injecting the pathogen through the cuticle of the tick. In addition to requiring live animals to feed ticks after the infection, this invasive method also has the disadvantage of a low survival rate of ticks after injection [95]. Nevertheless, its use by some authors has led to important results concerning *Babesia* spp. In 2018, Antunes and co-workers used a *R. bursa-B. ovis*-sheep infection model to characterize tick salivary gland genes that were differentially expressed in response to blood feeding and *B. ovis* infection [96]. In that experiment, female ticks were inoculated with *B. ovis* in the trochanter-coxae articulation and allowed to feed on rabbits. Vector competence was then confirmed by feeding *B. ovis*-infected ticks on a naïve lamb. This study allowed both increased understanding of the role of tick salivary gland genes in *Babesia* infections and identification of potential candidate vaccine antigens for innovative control strategies.

## 4. Conclusions

It is likely that vector competence for *Babesia* spp. has yet to be determined in some tick species. Furthermore, our understanding of the life cycles of *Babesia* spp. is still incomplete, especially regarding the intimate mechanisms of molecular dialogue between the parasite and its vertebrate and invertebrate hosts. Field experiments are less easy to control than those under laboratory conditions, and the gaps in our knowledge can probably only be filled by experimentally reproducing the transmission cycle as closely as possible to reality and by involving all three actors of the vectorial system. Indeed, this review has shown that the use of experimental systems for tick-borne pathogen infections that permit the complete transmission cycles of *Babesia* parasites has led to major scientific advances in the study of these pathogens. Despite these successes, efforts are still needed to standardize and simplify laboratory protocols to improve our ability to exploit tick artificial infection systems. It is hoped that, in the future, such models of artificial infection will be further developed in order to acquire new knowledge and develop new control strategies while avoiding the use of animals.

## Figures and Tables

**Figure 1 pathogens-10-01403-f001:**
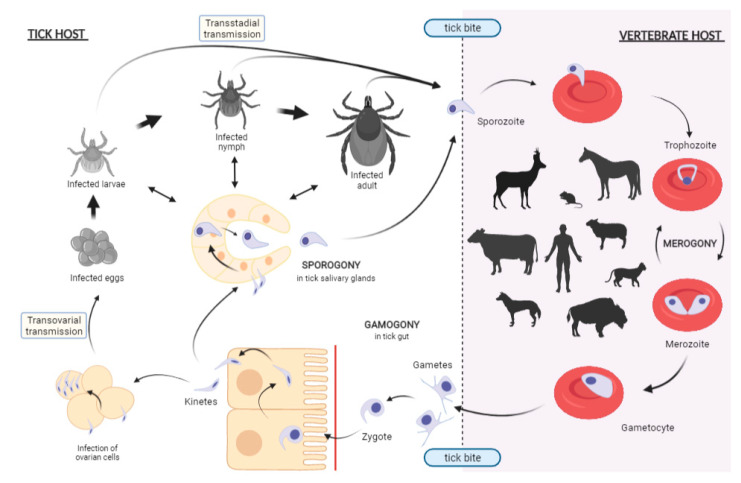
Life cycle of *Babesia* spp. sensu stricto. Vertebrate hosts are infected following the bite of an infected hard tick, through the invasion of the host erythrocytes by sporozoites excreted in the tick saliva. Inside the erythrocyte, sporozoites develop into trophozoites that undergo an asexual multiplication called merogony, ending in the formation of either merozoites that can infect other erythrocytes, or of gametocytes, which eventually develop into gametes. Infected erythrocytes are taken up by the tick during its blood meal, but only the gametocytes survive, and they then undergo further development, changing into gametes in the tick midgut. Then, sexual multiplication—gamogony—takes place with the fusion of two gametes to form a motile zygote that enters the midgut epithelial cells to develop into motile kinetes through meiotic division. Kinetes disseminate to tick tissues, including ovarian and salivary gland cells. The invasion of tick ovaries results in transovarial transmission while those of salivary glands leads to transmission to the vertebrate host through injection of sporozoites with the saliva. *Babesia* microti-like species, which belong to *Babesia* spp. s.l. species, only invade the salivary glands, not the ovary. The schematic representation was made using the software biorender.com.

**Figure 2 pathogens-10-01403-f002:**
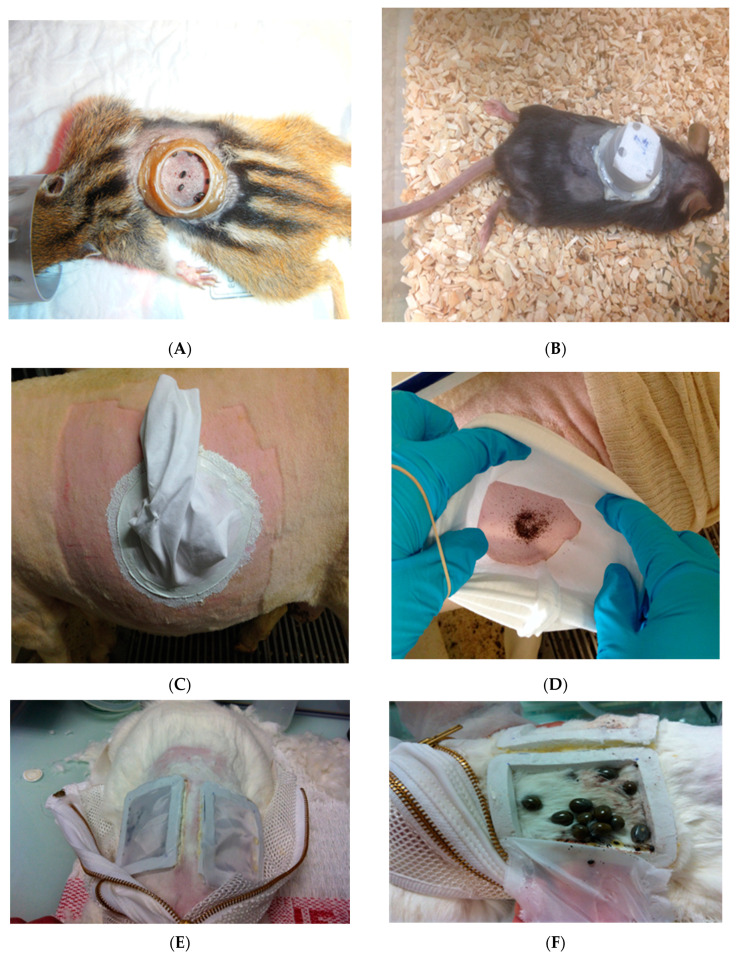
Experimental feeding of *Ixodes ricinus* ticks on (**A**) Siberian chipmunks (*Tamias sibiricus barberi)* [65], (**B**) mouse [66], (**C**,**D**) sheep [67], (**E**,**F**) rabbit [68], and (**G**,**H**) a membrane artificial feeding system [35].

**Table 1 pathogens-10-01403-t001:** Major *Babesia* species infecting humans, dogs, cattle, sheep, goat, and equids; their suspected or confirmed main vector and vertebrate hosts in the field; and the realization of the whole transmission cycle under experimental conditions.

*Babesia* spp.	Suspected or Confirmed Main Vectors	Main Vertebrate Hosts	Realization of the Complete Transmission Cycle Under Experimental Conditions
*Babesia microti*	*Ixodes persulcatus* *Ixodes ovatus* *Ixodes scapularis* *Ixodes trianguliceps* *Ixodes dammini* *Ixodes ricinus* *Rhipicephalus haemaphysaloides* *Haemaphysalis longicornis*	Human, rodent	ND ND [28] [29] [30] [31,32] [33] [34]
*Babesia divergens*	*Ixodes ricinus*	Human, cattle	[35,36,37,38,39]
*Babesia venatorum*	*Ixodes ricinus*	Human, roe deer	[40]
*Babesia duncani*	*Dermacentor albipictus*	Human, mule deer	ND
*Babesia bovis*	*Rhipicephalus microplus* *Rhipicephalus annulatus* *Rhipicephalus geigyi*	Cattle, buffalo	[41] ND ND
*Babesia bigemina*	*Rhipicephalus* *annulatus* *Rhipicephalus microplus* *Rhipicephalus decoloratus* *Rhipicephalus geigyi* *Rhipicephalus evertsi*	Cattle, buffalo	[42] [43,44] [45,46,47] ND ND
*Babesia major*	*Haemaphysalis punctata*	Cattle	[48,49]
*Babesia ovata*	*Haemaphysalis longicornis*	Cattle	[50]
*Babesia orientalis*	*Rhipicephalus haemaphysaloides*	Water buffalo	ND
*Babesia caballi*	*Dermacentor nitens* *Dermacentor sp.* *Hyalomma sp.* *Rhipicephalus evertsi*	Horse, Donkey, Mule	[51,52] ND ND [53]
*Babesia ovis*	*Rhipicephalus bursa*	Sheep and Goat	[54]
*Babesia motasi*	*Rhipicephalus bursa* *Haemaphysalis punctata*	Sheep and Goat	ND ND
*Babesia canis*	*Dermacentor reticulatus* *Haemaphysalis spp.* *Hyalomma spp.*	Dog	[55,56,57,58,59] ND ND
*Babesia gibsoni*	*Haemaphysalis sp.* *Rhipicephalus sanguineus*	Dog	ND ND
*Babesia vogeli*	*Rhipicephalus sanguineus*	Dog	ND
*Babesia rossi*	*Haemaphysalis* *elliptica* *Rhipicephalus sanguineus*	Dog	[60] ND

ND: no identified data.

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
