# Peer review of "Experimental Infection of Ticks: An Essential Tool for the Analysis of Babesia Species Biology and Transmission"

_pathogens, 2021, doi:10.3390/pathogens10111403_

Round 1
Reviewer 1 Report
The review is of interest but it is not comprehensive and several key references in the field covered by the study are missing. Specifically, the authors are exploring alternative tools for studying transmission of Babesia but neglected the discussion of in vitro sexual stage induction systems, which are described in several reports. In addition, other studies leading to the identification of proteins expressed exclusively in tick stages of Babesia are not mentioned. Much of these work also involve the molecular and morphological characterization of Babesia parasites in infected ticks. Together, these experimental approaches allowed for the identification of important molecules expressed by Babesia parasites in the tick stages, and the future development of transmission blocking vaccines against cattle Babesia parasites.
There is a large body of work revealing the usefulness of these above mentioned experimental approaches, which should be somehow acknowledged in a review paper proposing that: “Understanding the intimate mechanisms of molecular dialogue between the parasite and its vertebrate and in vertebrate hosts can only be achieved by experimentally reproducing the transmission cycle as closely as possible to reality and by involving the three actors of the vectorial system” as claimed by the authors in lines 120-123.
Also, one thing that needs to be emphasize by the authors is that humans are considered as “accidental” hosts in the natural history of these parasites. Therefore, and although Babesia parasites are able to establish infections, humans usually don’t play a significant role in the life cycle of the parasites and humans do not really contribute to their persistence in the environment.
Minor points for improvement:
Line 10: consistency with the use of capital letters (Canidae vs canidae?
Line 11: …is considered…rather than are
Line 12: is rather than are
Line 33: the use of babesicidal drugs should also be mentioned since it is an important tool for control of acute disease.
Lines 42, 43, 58: while or whereas rather than when
Line 73: this statement is not clear, the reference provided contradicts this statement. It is possible also that culture-adapted parasites may lose their transmission phenotype.
The life cycle should be clarified in several areas: 1] authors need to emphasize the differences in the cycle among sensu stricto and sensu lato (ie: B. microti) Babesias parasites that may have transovarial or transstadial modes of transmission, respectively.
It should also be pointed out that Rhipicephalus ticks such as R. microplus, which are the vectors of B. bovis and B. bigemina are single host ticks, so the whole life cycle of the ticks occurs in the same animal.
Author Response
First, we would like to thank the reviewer for his/her very constructive and helpful comments that improved the manuscript.
The review is of interest but it is not comprehensive and several key references in the field covered by the study are missing. Specifically, the authors are exploring alternative tools for studying transmission of Babesia but neglected the discussion of in vitro sexual stage induction systems, which are described in several reports. In addition, other studies leading to the identification of proteins expressed exclusively in tick stages of Babesia are not mentioned. Much of these work also involve the molecular and morphological characterization of Babesia parasites in infected ticks. Together, these experimental approaches allowed for the identification of important molecules expressed by Babesia parasites in the tick stages, and the future development of transmission blocking vaccines against cattle Babesia parasites.
There is a large body of work revealing the usefulness of these above mentioned experimental approaches, which should be somehow acknowledged in a review paper proposing that: “Understanding the intimate mechanisms of molecular dialogue between the parasite and its vertebrate and in vertebrate hosts can only be achieved by experimentally reproducing the transmission cycle as closely as possible to reality and by involving the three actors of the vectorial system” as claimed by the authors in lines 120-123.
We fully agree with the reviewer on the relevance of in vitro methods for characterization of the sexual stages of Babesia sp. and the identification of important molecules expressed by Babesia parasites in the tick stages. Some of them are listed in the introduction as examples : (i.e.: Maeda, H et al Initial development of Babesia ovata in the tick midgut. Vet Parasitol 2017; Bhat, et al. The invasion and growth of Babesia bovis in tick tissue culture. Experientia 1979.; Ribeiro et al: in vitro multiplication of sporokinetes in Ixodes scapularis (IDE8) cells. Exp Parasitol 2009; de Rezende et al In vitro cultivation and cryopreservation of Babesia bigemina sporokinetes in hemocytes of Rhipicephalus microplus. Vet Parasitol 2015)
However, the subject of the review, as stated in the title, abstract and introduction, concerns methods for the complete realization of tick transmission cycles under experimental conditions. And for us this does not include in vitro studies of gametocytogenesis induction nor in vitro studies of interaction between parasite and tick cells. The whole MS was revised in that sense and we hope that no confusion remains.
We agree that some references concerning the realization of the complete cycles in experimental conditions were missing in the first version of the MS, we apologize for that, and they have been added in the new version of the MS.
Also, one thing that needs to be emphasize by the authors is that humans are considered as “accidental” hosts in the natural history of these parasites. Therefore, and although Babesia parasites are able to establish infections, humans usually don’t play a significant role in the life cycle of the parasites and humans do not really contribute to their persistence in the environment.
This was corrected line 28 in the new version “The disease, impacting mainly cattle, sheep, goat, equidae, and canidae (and accidentally humans),…”
Minor points for improvement:
Line 10: consistency with the use of capital letters (Canidae vs canidae?
This was corrected
Line 11: …is considered…rather than are
This was corrected
Line 12: is rather than are
Is this the same correction requested above?
Line 33: the use of babesicidal drugs should also be mentioned since it is an important tool for control of acute disease.
This sentence was removed in the new version and the babesicidal drugs are mentioned now in line 64.
Lines 42, 43, 58: while or whereas rather than when
This was corrected
Line 73: this statement is not clear, the reference provided contradicts this statement. It is possible also that culture-adapted parasites may lose their transmission phenotype.
That’s effectively the idea we wanted to mention. The sentence was changed and we hope it is clearer this way: “Finally, it is important to mention that despite the promise of in vitro culture systems [14], maintenance by vitro culture [15] or needle-passage in the vertebrate host [16] in the absence of tick passage may generate significant changes in the parasite population, potentially creating a bias in research results.”
The life cycle should be clarified in several areas: 1] authors need to emphasize the differences in the cycle among sensu stricto and sensu lato (ie: B. microti) Babesias parasites that may have transovarial or transstadial modes of transmission, respectively.
Thanks for having highlighted this point, this was specified in the new version of the manuscript line 117: “The general life cycle of Babesia species is summarized in Figure 1 for Babesia sensu stricto (s.s.) species. Indeed, it is necessary to specify here that, quite recently, molecular phylogeny studies using the 18S rRNA gene have led to the division of Babesia species into two large groups: Babesia sensu stricto and Babesia sensu lato (s.l.). Species belonging to the latter group, such as B. microti, are not capable of transovarial transmission within the tick but only a transstadial mode of transmission [62,63].
It should also be pointed out that Rhipicephalus ticks such as R. microplus, which are the vectors of B. bovis and B. bigemina are single host ticks, so the whole life cycle of the ticks occurs in the same animal.
Thanks for that comment, this was specified in the new version of the manuscript lines 109-114: Most of the tick species listed as confirmed or suspected vectors of some Babesia species are three-hosts ticks - meaning that they take their three blood meals on three different hosts-. Some of them, however, have a two-host cycle such as some Hyalomma spp., Rhipicephalus evertsi and Rhipicephalus bursa, whereas Dermacentor nitens, Rhipicephalus annulatus, Rhipicephalus microplus and Rhipicephalus decoloratus are one-host ticks that use the same individual host animal for all active tick stages.
Reviewer 2 Report
More graphical images about Babesia distribution, host-ticks cycle/models or artificial feeding systems are need as a review, not just one table.
The author did not give enough their own vision for outlook. Even the opinion of lacking for current methods or their new ideas how to improve those methods.
As a review, such as Babesia orientalis, a pathogen can infect water buffalo should not missing in the manuscript.
Author Response
Comments and Suggestions for Authors
First, we would like to thank the reviewer for his/her constructive and helpful comments that improve the manuscript.
More graphical images about Babesia distribution, host-ticks cycle/models or artificial feeding systems are need as a review, not just one table.
Thank you for this comment. As requested, 2 figures were added in the new version: one representing the development cycle of Babesia and another one representing different models of tick feeding on both animal and artificial system.
The author did not give enough their own vision for outlook. Even the opinion of lacking for current methods or their new ideas how to improve those methods.
We have tried to improve this point in the new version and hope that it is now more satisfactory.
As a review, such as Babesia orientalis, a pathogen can infect water buffalo should not missing in the manuscript.
Babesia orientalis was added in the new version of the table 1 of the MS
Reviewer 3 Report
The submitted paper is of interest, well written and comprehensive.
However, Babesia canis subchapter needs more attention, to follow the same structure as the other Babesia spp. In this case, in focus is the protection against ticks and not the Babesia canis.
References need to be corrected, there are minor correction which should be corrected before further process.
e.g. line 493. It should be Parasitology Research
Author Response
First, we would like to thank the reviewer for his/her constructive and helpful comments that improve the manuscript.
Comments and Suggestions for Authors
The submitted paper is of interest, well written and comprehensive.
Thanks for this positive comment
However, Babesia canis subchapter needs more attention, to follow the same structure as the other Babesia spp. In this case, in focus is the protection against ticks and not the Babesia canis.
All of this part having been re-written, we hope that it is now more suitable
References need to be corrected, there are minor correction which should be corrected before further process.
e.g. line 493. It should be Parasitology Research
All bibliographic references have been reviewed and we hope that there are no more errors
Reviewer 4 Report
This review focuses on the experimental infection of ticks with Babesia, either by feeding ticks on experimentally infected animals, or by artificial tick feeding systems, but also on models to study the transmission of Babesia species. The authors summarized selected studies in separate sections, each section dealing with a single Babesia species.
General comments:
The English language and style will require extensive editing (the use of prepositions is particularly problematic) by a native English speaker.
The introduction and description of the life cycle (typo: 'like' cycle, L91) could be shortened. In the latter section, current knowledge on the phylogeny of Babesia spp. (e.g., Jalovecka et al., 2019. Babesia Life Cycle - When Phylogeny Meets Biology. Trends Parasitol. 35(5):356-368) could be integrated to briefly highlight differences in the life cycle of some Babesia spp., e.g. B. microti compared to the Babesia sensu stricto group.
Table 1 seems to be incomplete as some relevant papers were not cited. This for instance includes the vector role of I. ricinus for B. microti (Gray et al., 2002. Transmission studies of Babesia microti in Ixodes ricinus ticks and gerbils. J Clin Microbiol. 40(4):1259-63 - could also be discussed in section 3.5) and H. laechii as a probably vector for Babesia rossi (Kamani 2021. Molecular evidence indicts Haemaphysalis leachi (Acari: Ixodidae) as the vector of Babesia rossi in dogs in Nigeria, West Africa. Ticks Tick Borne Dis. 12(4):101717).
The level of detail in which the reviewed studies are described differ considerably and is not always comprehensible. I think that this review could benefit from an additional section before 3.1 describing and discussing the main outline of studies in which ticks are infected by feeding on infected animals (selection of animals, relevance of splenectomy, storage of Babesia strains, how are animals infected (needle inoculation vs. tick), containment of ticks when feeding on animals etc.).
An alternative infection method of ticks with Babesia is by the injection of ticks, as for instance shown by Battsetseg et al., 2007 (Babesia parasites develop and are transmitted by the non-vector soft tick Ornithodoros moubata (Acari: Argasidae). Parasitology. 134(Pt 1):1-8), but also in the cited study by Antunes et al (Antunes et al., 2018. Rhipicephalus bursa Sialotranscriptomic Response to Blood Feeding and Babesia ovis Infection: Identification of Candidate Protective Antigens. Front Cell Infect Microbiol. 2018 8:116). The authors should consider presenting and discussing this method separately as well.
In Section 4, artificial feeding methods using silicone membranes, e.g. Kröber & Guerin, 2007. In vitro feeding assays for hard ticks. Trends Parasitol. 2007 23(9):445-9 could be mentioned. Although they haven't been used for the artificial infection of ticks with Babesia spp., they have found wide deployment and use for infections with other tick-borne pathogens, e.g. Oliver et al., 2016. Infection of Immature Ixodes scapularis (Acari: Ixodidae) by Membrane Feeding. J Med Entomol. 53(2):409-15; Fourie et al., 2019. Transmission of Anaplasma phagocytophilum (Foggie, 1949) by Ixodes ricinus (Linnaeus, 1758) ticks feeding on dogs and artificial membranes. Parasit Vectors. 12(1):136) and may be used for establishing Babesia infection models in the future. The authors could also elaborate this section by discussing which Babesia spp. have been cultured in erythrocytes in vitro and may therefore be likely candidates to be used in such studies in the future.
Specific comments:
L26: delete (or explain) piroplasmosis, as this includes Theileria species as well
L26: why is Babesia considered on the rise? Is there a clear increase in nunber of cases?
L35: vaccines against canine Babesia species are no longer on the market as far as I know.
L170: numbers under ten are usually written out
L269: I noticed that this is the only Babesia species for which the acquisition feeding process (infection of ticks) is not described at all, only transmission studies are presented.
L284: it may help to add that D. nitens is a one-host tick species.
L315: NexGard
References: italicize scientific species names
Author Response
Comments and Suggestions for Authors
This review focuses on the experimental infection of ticks with Babesia, either by feeding ticks on experimentally infected animals, or by artificial tick feeding systems, but also on models to study the transmission of Babesia species. The authors summarized selected studies in separate sections, each section dealing with a single Babesia species.
First, we would like to thank the reviewer for his/her very constructive and helpful comments that improve the manuscript.
General comments:
The English language and style will require extensive editing (the use of prepositions is particularly problematic) by a native English speaker.
English was checked by Jeremy Gray, one of the Guest Reviewers of the topic and native English speaker, and this new version will surely be more satisfactory.
one of the Guest Reviewers
The introduction and description of the life cycle (typo: 'like' cycle, L91) could be shortened.
Both introduction and description of the life cycle were reduced as required and mistake was corrected.
In the latter section, current knowledge on the phylogeny of Babesia spp. (e.g., Jalovecka et al., 2019. Babesia Life Cycle - When Phylogeny Meets Biology. Trends Parasitol. 35(5):356-368) could be integrated to briefly highlight differences in the life cycle of some Babesia spp., e.g. B. microti compared to the Babesia sensu stricto group.
Thanks for having highlighted this point, this was specified in the new version of the manuscript line 117: “The general life cycle of Babesia species is summarized in Figure 1 for Babesia sensu stricto (s.s.) species. Indeed, it is necessary to specify here that, quite recently, molecular phylogeny studies using the 18S rRNA gene have led to the division of Babesia species into two large groups: Babesia sensu stricto and Babesia sensu lato (s.l.). Species belonging to the latter group, such as B. microti, are not capable of transovarial transmission within the tick but only a transstadial mode of transmission [62,63].
Table 1 seems to be incomplete as some relevant papers were not cited. This for instance includes the vector role of I. ricinus for B. microti (Gray et al., 2002. Transmission studies of Babesia microti in Ixodes ricinus ticks and gerbils. J Clin Microbiol. 40(4):1259-63 - could also be discussed in section 3.5) and H. laechii as a probably vector for Babesia rossi (Kamani 2021. Molecular evidence indicts Haemaphysalis leachi (Acari: Ixodidae) as the vector of Babesia rossi in dogs in Nigeria, West Africa. Ticks Tick Borne Dis. 12(4):101717).
We apologize for the omissions of Gray et al., 2002. that should effectively be mentioned in the MS and was added in the new version. However, the recent study of Kamani and al. concerns molecular identification of parasite in progenies of field collected ticks and no experimental model of transmission so, for us, it should not be mentioned here.
The level of detail in which the reviewed studies are described differ considerably and is not always comprehensible. I think that this review could benefit from an additional section before 3.1 describing and discussing the main outline of studies in which ticks are infected by feeding on infected animals (selection of animals, relevance of splenectomy, storage of Babesia strains, how are animals infected (needle inoculation vs. tick), containment of ticks when feeding on animals etc.).
Thank you for this helpful advice. Since the entire manuscript has been reviewed and changed, we hope that this will be improved in this new version
An alternative infection method of ticks with Babesia is by the injection of ticks, as for instance shown by Battsetseg et al., 2007 (Babesia parasites develop and are transmitted by the non-vector soft tick Ornithodoros moubata (Acari: Argasidae). Parasitology. 134(Pt 1):1-8), but also in the cited study by Antunes et al (Antunes et al., 2018. Rhipicephalus bursa Sialotranscriptomic Response to Blood Feeding and Babesia ovis Infection: Identification of Candidate Protective Antigens. Front Cell Infect Microbiol. 2018 8:116). The authors should consider presenting and discussing this method separately as well.
Thank you for that remark. This was indeed an error that was corrected and the new version includes a section dedicated to this tick infection method with the reference of Antunes et al. However, the reference of Battsetseg et al, despite being very interesting, concerns Theileria (Babesia) equi and because of its belonging to the genus Theileria, we have chosen not to mention this species in the whole manuscript which therefore only concerns the parasites from the Babesia genera.
In Section 4, artificial feeding methods using silicone membranes, e.g. Kröber & Guerin, 2007. In vitro feeding assays for hard ticks. Trends Parasitol. 2007 23(9):445-9 could be mentioned. Although they haven't been used for the artificial infection of ticks with Babesia spp., they have found wide deployment and use for infections with other tick-borne pathogens, e.g. Oliver et al., 2016. Infection of Immature Ixodes scapularis (Acari: Ixodidae) by Membrane Feeding. J Med Entomol. 53(2):409-15; Fourie et al., 2019. Transmission of Anaplasma phagocytophilum (Foggie, 1949) by Ixodes ricinus (Linnaeus, 1758) ticks feeding on dogs and artificial membranes. Parasit Vectors. 12(1):136) and may be used for establishing Babesia infection models in the future. The authors could also elaborate this section by discussing which Babesia spp. have been cultured in erythrocytes in vitro and may therefore be likely candidates to be used in such studies in the future.
Membrane feeding techniques are much more presented and discussed in the new version of the manuscript. However, it is difficult to present here all the references that used these techniques to infect ticks with different pathogens other than Babesia or to mention some and not others. As the present review focuses on Babesia, the choice was then made here to discuss only studies on Babesia infection and to mention a published review that mentions the other ones (Bonnet et al, 2012) and we hope this will be appropriate.
Specific comments:
L26: delete (or explain) piroplasmosis, as this includes Theileria species as well
Piroplasmosis was deleted in the new version
L26: why is Babesia considered on the rise? Is there a clear increase in nunber of cases?
Some authors say it but as it is true that there is no number to validate this information at a global level, "on the rise" has been removed from the new version
L35: vaccines against canine Babesia species are no longer on the market as far as I know.
This information has been removed in the new version
L170: numbers under ten are usually written out
This was corrected
L269: I noticed that this is the only Babesia species for which the acquisition feeding process (infection of ticks) is not described at all, only transmission studies are presented.= B canis
All of this part having been re-written, we hope that it is now more suitable
L284: it may help to add that D. nitens is a one-host tick species.
This was added in the new version lines 109-114: Most of the tick species listed as confirmed or suspected vectors of some Babesia species are three-hosts ticks - meaning that they take their three blood meals on three different hosts-. Some of them, however, have a two-host cycle such as some Hyalomma spp., Rhipicephalus evertsi and Rhipicephalus bursa, whereas Dermacentor nitens, Rhipicephalus annulatus, Rhipicephalus microplus and Rhipicephalus decoloratus are one-host ticks that use the same individual host animal for all active tick stages.
L315: NexGard
This was corrected
References: italicize scientific species names
All bibliographic references have been reviewed and we hope that there are no more errors
Round 2
Reviewer 1 Report
The manuscript have been improved. However, I suggest that the authors should address the following points:
- The authors mention the existence of “important gaps in knowledge”. This is a very general statement. It would help readers to pinpoint what are those gaps, or at least the more important ones, so they can be prioritized.
- Is it really necessary to have a full understanding of the 3-part relationship in order to have control measures? In any case what type and level of knowledge would still be required?
- Line 52: New agents of piroplasmosis, such as T. hayenii have been recently identified, in addition to B. caballi and T. equi.
- Line 76: use “animals” rather than “animal”
- Line 88: use. molecular interactions
- Table 1: Haemaphysalis longicornis has also been identified as a vector for B. microti. This info should be added to Table 1. Please check the following reference: Identification of Haemaphysalis longicornis Genes Differentially Expressed in Response to Babesia microti Infection May 2020Pathogens 9(5):378. DOI: 10.3390/pathogens9050378 ]
- Line 139: Still needs to be clarified that at least for one host ticks the fact that kinetes infect salivary glands of the primary tick is irrelevant, since these ticks are unable to transmit via transstadial mechanisms. However, kinetes should reach and invade the salivary glands of the next generation larva during TO transmission.
- Line 169 (legend) you may use “…membrane artificial feeding system”
- The title of section 3.1 should be “Tick infestations on Babesia spp infected animals” . Infestation rather than infection should be used for ticks and other ectoparasites.
- I suggest some rewriting in Lines 219-224. For instance, “…it allowed to demonstrate…” should be replaced by “it allowed demonstrating”.
Author Response
The manuscript have been improved. However, I suggest that the authors should address the following points:
Another time, many thanks for the thoughtful review of our manuscript and constructive advices
- The authors mention the existence of “important gaps in knowledge”. This is a very general statement. It would help readers to pinpoint what are those gaps, or at least the more important ones, so they can be prioritized.
The following sentence was added line 67-68 in the new version, and we hope this meets the reviewer’s request: “The development of improved control measures against babesiosis is limited by the numerous and significant gaps in our understanding of the biology of Babesia spp., especially regarding molecular interaction between parasites, vectors and vertebrate hosts, as well as the factors that may influence both the development and the transmission of the parasite [11].”
- Is it really necessary to have a full understanding of the 3-part relationship in order to have control measures? In any case what type and level of knowledge would still be required?
We hope that the changes made above also respond to this comment as it implies the study of the interactions between the parasite and the vertebrate host, the vertebrate host and the tick and the parasite and the tick; each step can be targeted to interrupt the development/transmission of the parasite
- Line 52: New agents of piroplasmosis, such as T. hayenii have been recently identified, in addition to B. caballi and T. equi.
The reviewer wants probably to talk about T. haneyi? (Knowles DP, Kappmeyer LS, Haney D, Herndon DR, Fry LM, Munro JB, Sears K, Ueti MW, Wise LN, Silva M, Schneider DA, Grause J, White SN, Tretina K, Bishop RP, Odongo DO, Pelzel-McCluskey AM, Scoles GA, Mealey RH, Silva JC (2018) Discovery of a novel species, Theileria haneyi n. sp., infective to equids, highlights exceptional genomic diversity within the genus Theileria: implications for apicomplexan parasite surveillance. Int J Parasitol 48:679–690), according to that it was added in the new version of the MS ligne 53: “In equids, Babesia caballi is (with Theileria equi and T. haneyi) the agent of equine piroplasmosis…” with the corresponding reference.
- Line 76: use “animals” rather than “animal”
This was corrected
- Line 88: use. molecular interactions
This was corrected
- Table 1: Haemaphysalis longicornis has also been identified as a vector for B. microti. This info should be added to Table 1. Please check the following reference: Identification of Haemaphysalis longicornis Genes Differentially Expressed in Response to Babesia microti Infection May 2020Pathogens 9(5):378. DOI: 10.3390/pathogens9050378 ]
Sorry for that oversight. In the reference mentioned, the entire transmission cycle was not carried out but on the other hand it was carried out in the following paper which was therefore added to the MS: Wu, J.; Cao, J.; Zhou, Y.; Zhang, H.; Gong, H.; Zhou, J. Evaluation on Infectivity of Babesia microti to Domestic Animals and Ticks Outside the Ixodes Genus. Front Microbiol 2017, 8, 1915, doi:10.3389/fmicb.2017.01915.
- Line 139: Still needs to be clarified that at least for one host ticks the fact that kinetes infect salivary glands of the primary tick is irrelevant, since these ticks are unable to transmit via transstadial mechanisms. However, kinetes should reach and invade the salivary glands of the next generation larva during TO transmission.
According to that comment, the following sentence was added to the new version: “The ticks thus transmit the sporozoites to a new host during a new blood meal of the next life-stage for the ticks with several hosts or of the next generation after transovarian transmission for the one-host ticks.”
- Line 169 (legend) you may use “…membrane artificial feeding system”
This was corrected
- The title of section 3.1 should be “Tick infestations on Babesia spp infected animals” . Infestation rather than infection should be used for ticks and other ectoparasites.
This was corrected
- I suggest some rewriting in Lines 219-224. For instance, “…it allowed to demonstrate…” should be replaced by “it allowed demonstrating”.
This was corrected
Reviewer 2 Report
The authors fixed all the comments, it seems much better as a review now.
Author Response
The authors fixed all the comments, it seems much better as a review now.
Thanks for that positive feed back
Reviewer 4 Report
The revised vresion of this manuscript has greatly improved, what follows are minor comments and suggestions:
L57, L60, L159, L278: write out genus names at the start of a sentence
L82: and animal models
L89: control methods
L90: delete 'these'
Tabel 1: Rhipicephalus sanguineus s.l. is not a vector for B. canis, references 53 and 54 are not appropriate here. This tick is a vector for B. vogeli, see Penzhorn BL. Don't let sleeping dogs lie: unravelling the identity and taxonomy of Babesia canis, Babesia rossi and Babesia vogeli. Parasit Vectors. 2020 13(1):184 and possibly Brumpt E. Transmission de la piroplasmose canine tunisienne par le Rhipicephalus sanguineus. Bull Soc Pathol Exot. 1919;12:651–4 for details.
L191: italicize R. microplus
L207: pathogen or pathogen DNA alone
L220: live animals
L223: Babesia (capital B)
L292: to limit the use of animals as much as possible
L294: can be expensive and difficult or even impossible in the case of
L297: whole feeding period of the ticks is
L313: in the absence of blood - but this is not always the case, as the following example from Inokuma & Kemp shows. Suggest to omit 'in the absence of blood'
Author Response
The revised vresion of this manuscript has greatly improved, what follows are minor comments and suggestions:
Another time, many thanks for the thoughtful review of our manuscript and constructive advices
L57, L60, L159, L278: write out genus names at the start of a sentence
This was corrected
L82: and animal models
This was corrected
L89: control methods
This was corrected
L90: delete 'these'
This was corrected
Tabel 1: Rhipicephalus sanguineus s.l. is not a vector for B. canis, references 53 and 54 are not appropriate here. This tick is a vector for B. vogeli, see Penzhorn BL. Don't let sleeping dogs lie: unravelling the identity and taxonomy of Babesia canis, Babesia rossi and Babesia vogeli. Parasit Vectors. 2020 13(1):184 and possibly Brumpt E. Transmission de la piroplasmose canine tunisienne par le Rhipicephalus sanguineus. Bull Soc Pathol Exot. 1919;12:651–4 for details.
This is indeed an error with references that have nothing to do here, thank you for noting it and this has been corrected in the new version
L191: italicize R. microplus
This was corrected
L207: pathogen or pathogen DNA alone
This was corrected
L220: live animals
This was corrected
L223: Babesia (capital B)
This was corrected
L292: to limit the use of animals as much as possible
This was corrected
L294: can be expensive and difficult or even impossible in the case of
This was corrected
L297: whole feeding period of the ticks is
This was corrected
L313: in the absence of blood - but this is not always the case, as the following example from Inokuma & Kemp shows. Suggest to omit 'in the absence of blood'
This has been replaced with the following wording and we hope this will be appropriate:
“and this, regardless of the "true" blood meal.»